

# Leveraging PSO-MLP for intelligent assessment of student learning in remote environments: a multimodal approach

Jing Wang[1] and Muhammad Asif[2]

[1] College of Education Science, Xin Jiang Normal University, Urumqi, Xinjiang, China
[2] Department of Computer Science, National Textile University, Faisalabad, Pakistan

## ABSTRACT

The rapid advancement of artificial intelligence (AI) has catalyzed transformative changes in education, particularly in mobile and online learning environments. While existing deep learning models struggle to efficiently integrate the complexity of remote education data and optimize model performance, this article proposes an intelligent evaluation method for students' learning states based on multimodal data. First, the joint characteristics of the pre-class mental status survey information and the health big data of teachers and students in the online teaching process constitute input data. Then, the multilayer perceptron (MLP) is used to intelligently identify the students' status and classify their enthusiasm for the class. Finally, the particle swarm optimization (PSO) model is used to optimize the model and improve the overall recognition rate. Compared to traditional methods, the PSO-MLP model with combined multimodal data performs well, achieving an accuracy of 0.891. It provides an operational, technical solution for the education system, provides a new AI foundation for personalized teaching and student health management by accurately assessing students' learning status, and helps to improve the effectiveness and efficiency of remote education.

## INTRODUCTION

Education plays a pivotal role in shaping individuals' capacities to pursue happiness, realize aspirations, and contribute meaningfully to society. However, traditional educational methodologies are increasingly strained as they adapt to the rapid pace of technological advancements and societal changes. The concept of educational reform has thus emerged as a critical endeavor aimed at modernizing pedagogical practices to meet contemporary needs (*Chen, Zheng & Yu, 2020*; *Li et al., 2023*).

The rise of remote education, accelerated by the COVID-19 pandemic, has fundamentally transformed teaching and learning environments. While it has enabled broader access to educational resources, this shift has also revealed persistent challenges, particularly in maintaining student engagement and assessing learning outcomes in the absence of physical presence. Although disparities in access to digital infrastructure and digital literacy remain significant barriers for some student populations (*Villegas-Ch et al., 2021*; *Liu et al., 2021*), a more pressing issue in technologically equipped settings is the lack

Corresponding author
Jing Wang, 17399056639@163.com

of real-time insight into students' cognitive and emotional states. Traditional online platforms often fail to capture learners' affect and attention, limiting instructors' ability to deliver adaptive and personalized support. This gap has motivated the development of intelligent assessment systems that leverage physiological signals to infer learner states, thereby enhancing engagement and learning effectiveness in remote settings. Beyond technological access, the prolonged use of electronic devices in online learning environments has been linked to adverse physical and mental health outcomes among students. Increased screen time contributes to visual strain, fatigue, and potential long-term health risks. At the same time, reduced physical activity and social interaction exacerbate mental health challenges such as social isolation and anxiety (*Wangang, 2023*). These multifaceted challenges underscore the urgent need for innovative methodologies that can comprehensively assess and mitigate the impacts of remote education on student well-being. In this context, the integration of advanced data analytics, particularly through big data frameworks, offers promising solutions. Big data analytics enables the real-time monitoring and analysis of vast datasets encompassing students' physical health metrics, mental states, and learning behaviors (*Dávila-Montero et al., 2021*). By leveraging multimodal data—such as heart rate variability and exercise patterns—from health, big data repositories, educators and health professionals can gain deeper insights into students' holistic well-being. This data-driven approach not only enhances the accuracy of assessing students' learning states but also provides actionable insights to inform personalized interventions and support mechanisms.

## Motivation for the study and methodology

The increasing prevalence of remote and mobile learning environments has introduced significant challenges in accurately assessing student engagement, emotional well-being, and learning effectiveness. Traditional evaluation methods, relying heavily on self-reporting or single-modal behavioral indicators, are often limited by subjectivity, latency, and low adaptability. This study is motivated by the urgent need to develop intelligent, real-time, and objective evaluation tools that can adapt to the multimodal and dynamic nature of distance education. The decision to integrate multimodal physiological signals—such as electroencephalography (EEG), galvanic skin response (GSR) and heart rate—with intelligent classification models is based on the understanding that physiological data serve as reliable proxies for emotional and cognitive states, which are essential for effective learning.

To address the inherent complexity and non-linearity of such data, a multilayer perceptron (MLP) was selected due to its strong capacity for modeling high-dimensional feature interactions. However, MLP's performance is highly sensitive to its initial hyperparameters and susceptible to local minima during training. Therefore, particle swarm optimization (PSO) was introduced as a metaheuristic method to efficiently optimize MLP parameters. PSO offers global search capabilities and faster convergence, making it well-suited for optimizing neural networks in scenarios involving heterogeneous, high-noise inputs. The combined PSO-MLP framework thus aligns with the study's

motivation to create a robust, adaptive, and interpretable model for intelligent assessment in real-world, data-rich educational settings.

Therefore, the contributions of the article are:

(1) Integration of multimodal data for enhanced student evaluation: The article introduces an intelligent evaluation method that integrates pre-class mental status surveys with big health data from both teachers and students. This multimodal approach enables a more comprehensive understanding of students' learning states and classroom enthusiasm, representing a significant improvement over existing deep learning models that struggle with remote education data.

(2) Application of particle swarm optimization (PSO) to improve model performance: By applying the PSO model to optimize the Multilayer Perceptron (MLP), the article enhances the efficiency and accuracy of the learning state assessment. The PSO-MLP model achieves a notable accuracy of 0.891, demonstrating improved performance over traditional methods and offering a technical solution for better recognizing and managing students' learning conditions.

'Related Works' analyzes the application status of the PSO-MLP model under multimodal data, and 'Materials and Methods' expounds on the implementation process of the intelligent evaluation method of students' learning status based on multimodal data. In 'Experiments and Analysis', the robustness of the model and the effect of sentiment classification are experimented with. 'Discussion' concludes with a summary and limitation analysis.

## RELATED WORKS

In distance education, researchers predict students' learning states based on their course engagement data. *Liu et al. (2025)* extract useful features automatically from course engagement data using a convolutional neural network with long short-term memory (LSTM) models to forecast learning states across different periods dynamically. *Jin (2023)* uses learning behavior data to employ a binary logistic regression model for predicting student dropout rates. *Shen et al. (2022)*, leveraging course data, extracting 19 features and constructing sliding window models using machine learning algorithms to predict learners' emotional trends dynamically.

However, relying solely on single-feature extraction methods leads to lower accuracy in identifying students' online learning states. Multimodal data analysis integrates various data types to provide a comprehensive view of student learning states, enhancing the perceptual capabilities of deep learning models. Analysis of written assignments and discussion forums can provide valuable insights into student comprehension and participation (*Xu, Chen & Chen, 2020*). Tracking interactions with learning management systems helps identify patterns of student behavior and engagement (*Veluvali & Surisetti, 2022*). Additionally, data from wearable devices, such as heart rate and skin conductance, offer additional dimensions on student engagement and stress levels (*Hernández-Mustieles et al., 2024*). These diverse data sources enable a more comprehensive understanding of student learning, thereby providing insights for more effective and personalized educational interventions. *Xu et al. (2021)* introduced the concept of bilinear pooling

primarily for feature fusion. The primary idea of bilinear pooling is to combine and cross-multiply two features to produce a fused feature. *Almujally et al. (2024)*, using a multimodal compact bilinear pooling method to fuse video and audio-related features, achieved an accuracy of 97%, demonstrating superior recognition performance compared to other algorithms.

PSO and MLP have been widely applied in educational research and practice due to their robust optimization and pattern recognition capabilities. MLP, an artificial neural network, excels in modeling complex relationships within data. In educational assessment, MLP is used to predict student success, identify learning barriers, and customize personalized learning experiences (*Smirani et al., 2022*). *Mandia, Mitharwal & Singh (2024)* demonstrated the effectiveness of MLP in simulating student engagement and performance, offering valuable insights for educators to design better teaching strategies. Combining PSO with MLP harnesses PSO's optimization strengths and MLP's predictive capabilities, thereby enhancing the performance of educational assessments. Research indicates that PSO can optimize neural network hyperparameters, such as learning rates, neuron numbers, and layers, thereby improving model accuracy and convergence speed (*Xiao et al., 2022*). In the educational domain, particle swarm algorithms are utilized to optimize the parameters of various educational algorithms and systems. For instance, *Kumari & Kumar (2023)* applied PSO to optimize neural network parameters for predicting student performance, achieving higher accuracy compared to traditional optimization methods.

Therefore, integrating PSO and MLP for intelligent assessment offers advantages such as high optimization efficiency, high prediction accuracy, and strong adaptability. PSO optimizes MLP hyperparameters, thereby enhancing model accuracy and accelerating convergence speed (*Al Bataineh & Manacek, 2022*). The PSO-MLP model is superior to traditional MLP models in predicting student grades and identifying at-risk students (*Putra et al., 2021*; *Huang et al., 2025*). Furthermore, the PSO-MLP model can adapt to various educational contexts by merging different types of data. Recent advancements in transformer-based models have shown substantial promise in the analysis of multimodal physiological signals. For example, *Wu, Daoudi & Amad (2023)*, *Lin et al. (2025)* introduced a transformer model designed for wearable-based emotion recognition using heart rate, skin conductance, and accelerometer data, achieving 71.5% accuracy in binary affect classification. Similarly, *Mordacq et al. (2024)* proposed ADAPT (Anchored Physiological Transformer), a masked multimodal transformer that demonstrated robust performance in predicting stress under high-G conditions. *Alazeb et al. (2024)* introduced VidFormer, a hybrid 3D-CNN and transformer framework for estimating heart rate and respiration from facial video, demonstrating strong generalization across subjects and lighting conditions. These works illustrate the growing impact of transformer architectures in learning complex interdependencies among physiological modalities, offering methodological insight and inspiration for future extensions of this study.

Despite these advantages, integrating PSO and MLP in educational assessment faces challenges and limitations. Integrating multimodal data (including personal and behavioral data) increases model computational complexity, requiring substantial

computational resources and efficient algorithms to handle large datasets. Achieving the PSO-MLP model in large-scale educational environments is challenging due to the need for robust infrastructure and real-time data processing capabilities. Moreover, integrating data from various sources (text, behavioral, and physiological) necessitates complex data integration techniques to ensure consistency and accuracy.

Multimodal data fusion strategies are another vital area of development, focusing on the effective combination of information from different modalities. Techniques such as concatenation, attention mechanisms, and tensor fusion have been proposed to integrate multimodal data while minimizing redundancy and maximizing complementary information (*Duan et al., 2024*; *Liu, Luo & Fu, 2024*). These strategies are crucial for enhancing the accuracy and robustness of PSO-MLP models without significantly increasing computational complexity, thereby contributing to more effective and efficient models.

Emerging research is also focusing on real-time and incremental learning methods for multimodal data, which enable PSO-MLP models to continuously update their parameters as new data becomes available (*Liu et al., 2023*; *Wang et al., 2025*; *Liu, Cao & Chen, 2024*). This approach reduces the need for retraining from scratch, thereby lowering computational costs and making the models more adaptable to dynamic environments and streaming data. This is particularly relevant for applications requiring real-time processing and decision-making.

# MATERIALS AND METHODS

## Data preprocessing steps

The accuracy and robustness of multimodal learning state recognition heavily depend on the quality and consistency of the input data. Therefore, an extensive data preprocessing pipeline was implemented to standardize the input from various physiological signal sources, including EEG, GSR, heart rate, and facial expression. The preprocessing procedures were designed to ensure data integrity, temporal alignment, and feature relevance, thereby facilitating effective learning by the PSO-MLP model.

### Data cleaning

Raw data acquired from the DEAP dataset contains noise and missing values due to sensor artifacts and environmental interference. To address this, the following cleaning steps were applied:

Outlier removal using z-score thresholding ($|z| > 3$).

Artifact correction through bandpass filtering for EEG signals (4–45 Hz).

Missing value imputation *via* linear interpolation to preserve temporal continuity.

### Signal segmentation and alignment

To facilitate multimodal fusion, all signals were segmented into non-overlapping windows of one-second duration. This ensured:

Temporal synchronization across modalities;

uniform sample length per instance;

preservation of temporal dependencies within each time window.

*Feature extraction*

Each modality underwent domain-specific feature extraction to convert raw signals into structured input features:

EEG features: power spectral density (PSD), band power (theta, alpha, beta), and Hjorth parameters.

GSR and heart rate features: time-domain metrics including mean, standard deviation, skewness, and kurtosis.

Facial expression features: geometric distances between key landmarks and action unit (AU) activation scores.

Sliding windows with a 50% overlap were used during feature extraction to enhance temporal resolution and capture dynamic changes.

*Feature normalization*

Due to the heterogeneous nature of the extracted features, normalization was applied:

Z-score normalization to enforce zero-mean and unit-variance across each feature.

Min-max scaling was employed in parallel experiments to assess the effect of the normalization strategy on model convergence.

*Label encoding and data augmentation*

To support supervised learning:

Labels were encoded into three emotional states (Active, Negative, and Exhausted) based on annotated DEAP scores.

Data augmentation was performed using generative adversarial networks (GANs) to synthetically generate balanced samples across underrepresented classes, mitigating class imbalance and improving generalization. To address the class imbalance and enhance the robustness of our model, we utilized a GAN to augment the EEG data. The implementation details are as follows:

**GAN architecture: Generator:** A feedforward neural network with three hidden layers (128, 256, and 512 neurons, respectively), using Leaky ReLU activation and tanh in the output layer. **Discriminator:** A mirrored architecture with dropout (0.3) added for regularization, ending with a sigmoid output for binary classification. **Input:** Random noise vector of size 100. **Training Configuration:** Optimizer: Adam with learning rate = 0.0002, β1 = 0.5 Batch size: 64 Epochs: 1000 Loss function: Binary cross-entropy Additionally, classification accuracy with and without synthetic data was compared to evaluate augmentation effectiveness.

## Remote monitoring

The underlying distribution of real data samples is captured through the GAN generator and new data is generated, which divides the original data into three groups according to the labels (high "2", medium "1", and low "0"). The objective function of GAN is expressed as follows.

$$\min_{G} \max_{D} \left\{ f(D, G) = E_{x \sim \text{Pdata}(x)}[\log D(x)] + E_{z \sim Pz(z)}[1 - D(G(z))] \right\} \tag{1}$$

where x fits the true data distribution $P_{data}(x)$, z fits the prior distribution $P_z(z)$, and $E(\cdot)$ represents the expected value. The Discriminator $D$ is a neural network designed to distinguish between real data samples and those generated by the Generator. The Generator $G$ is a neural network that generates synthetic data samples from a noise vector $z$ sampled from a prior distribution $P_z(z)$.

Therefore, the Wasserstein distance is introduced based on GAN, which effectively addresses the issue that the distance cannot be reflected when the two distributions have no overlap. The Wasserstein distance is expressed as follows.

$$W(p, q) = \inf_{\gamma \sim (p,q)} E_{(x,y) \sim \gamma}[\|x - y\|] \tag{2}$$

where $\inf\{\cdot\}$ set lower bounds, $\prod(p, q)$ is the distribution of all possible sets of joint distribution, joint distribution, for each possible $\gamma \sim \prod(p, q)$, calculate the expectation of distance $\|x - y\|$, called $E_{(x,y) \sim \gamma}[\|x - y\|]$. The lower bound on the expectation is the Wasserstein distance of the distributions p and q.

After the aforementioned preprocessing steps, we obtained the processed physiological signal S(t). Figure 1 demonstrates the preprocessing effects: Fig. 1A shows the raw data during the emotional stimulation phase; Fig. 1B displays the data after linear interpolation; Fig. 1C presents the data after noise reduction with a filter; Fig. 1D shows the normalized data; Fig. 1E depicts the data after processing with GAN. These techniques not only enhance the accuracy of emotion recognition but also provide crucial technical support and data foundation for applications in remote monitoring and education.

## MLP-based health condition recognition

MLP is the most classical neural network, also known as a feedforward neural network or artificial neural network. The traditional perceptron is a linear model that can only handle simple binary classifications and performs poorly in nonlinear problems. Then, by adding hidden layers and activation functions, it effectively improves the nonlinear expression ability and is widely used in data mining, machine learning, and other fields (*Xiong et al., 2021*). The specific structure of the MLP is shown in Fig. 2.

For a multilayer perceptron, its internal neural network layer is also divided into three types: input layer, hidden layer, and output layer. Corresponding weights connect different layers. Input $u = [u_1, u_2, \cdots, u_n]^T$ and $y = [y_1, y_2, \cdots, y_p]^T$ are mapped by the weight w shown in Formula (3) between each layer:

$$w = \left[ (w^1)^T, (w^2)^T \right]^T. \tag{3}$$

Once the MLP is constructed, the number of parameters can be calculated. The learning procedure for the MLP involves estimating the parameters. To optimize the MLP training, a general state-space model can be expressed by Formulas (4) and (5):

$$w_k = w_{k-1} + v_k \tag{4}$$

$$y_k = g(w_k, u_k) + \omega_k. \tag{5}$$

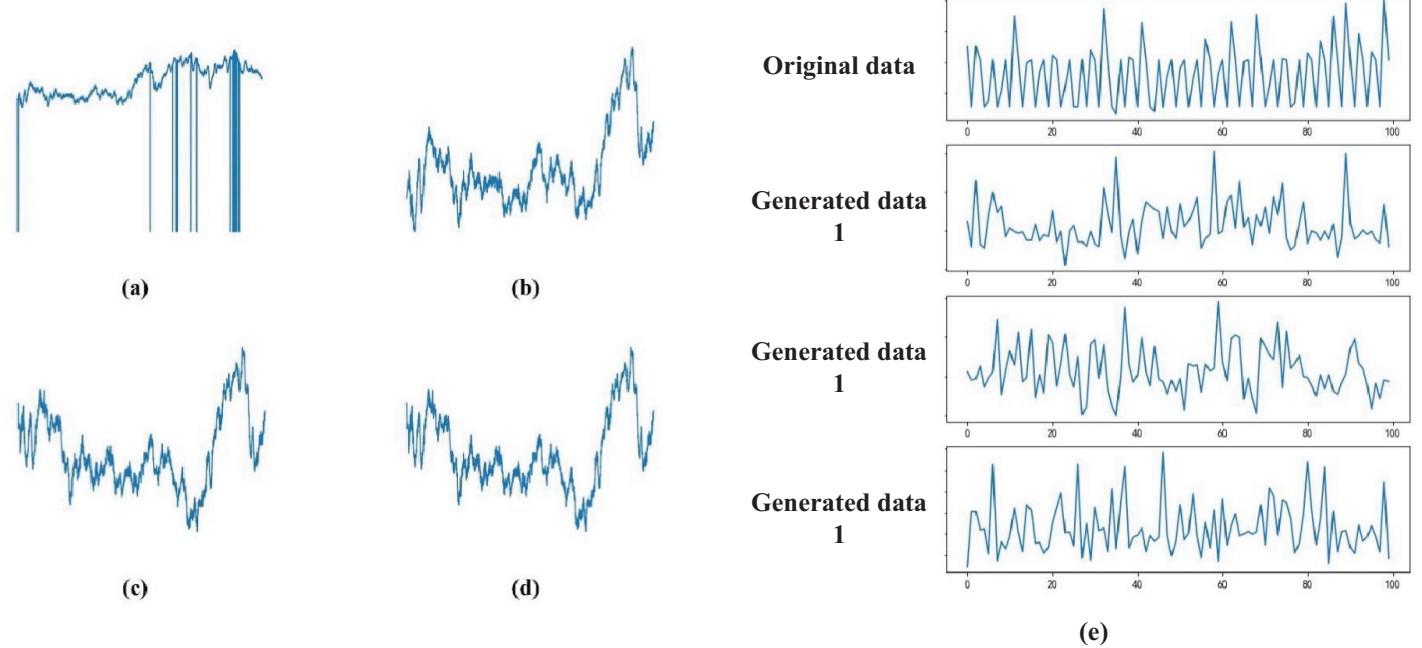

**Figure 1** (A–E) Data signal processing results.

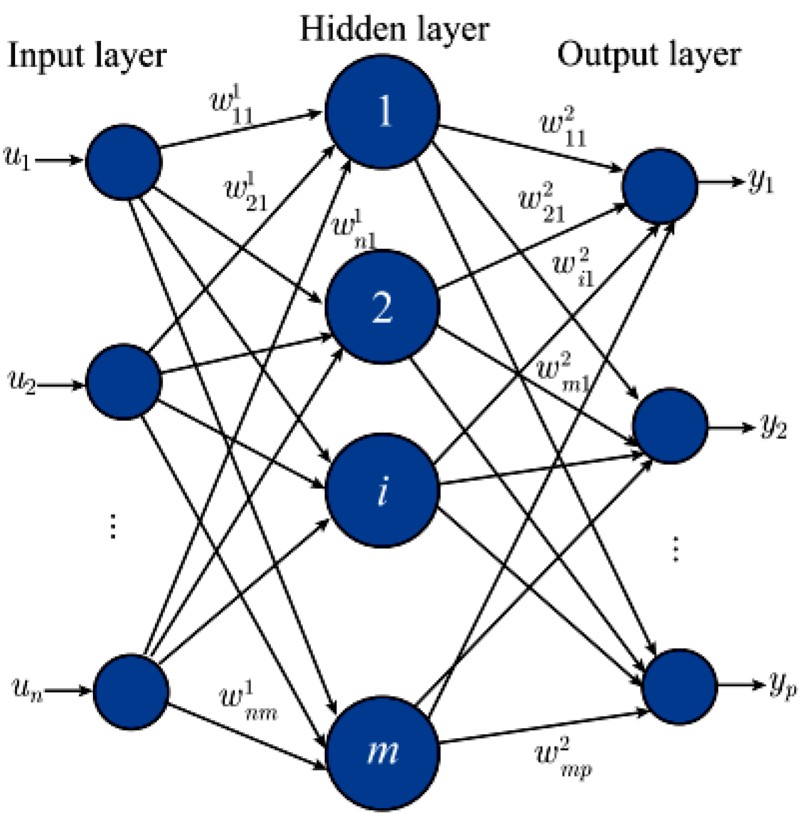

**Figure 2** Construction of the MLP.

For its learning process and parameter iteration, it can be calculated through the forward and backward propagation algorithm. The forward propagation algorithm is used to establish the input, output, and network connection modes, as well as to determine the number of layers and the corresponding nodes in the network. The forward propagation algorithm only initializes the neural network and does not adjust the weights and offsets. It only performs feedforward calculations and cannot solve problems. Therefore, we need to use the backpropagation method to adjust the internal parameters of the network and optimize its internal structure to minimize the defined loss function. The specific process of the reverse neural network is shown in Algorithm 1.

This algorithm represents the foundational process of training a backpropagation neural network, wherein the minimization of the loss function guides the iterative adjustment of weights and biases. The forward propagation computes activations, while backpropagation calculates the gradients, enabling the optimization of network parameters. The process iterates until the changes in parameters fall below a predefined threshold, ensuring convergence to an optimal solution.

## PSO-based model optimization

The neural network methods typically employ gradient descent for learning and optimizing network weights during model optimization, making them sensitive to initial values. Improper selection of initial values can cause optimal weights and biases to converge to local extreme points, significantly reducing model accuracy. Therefore, this study aims to utilize the PSO method to optimize the initial values of MLP, thereby enhancing performance. This article evaluates the performance of particles through their adoption in the study.

The particles in the particle swarm used for the MLP parameters optimization are defined as follows:

$$P_j = \lfloor C_j \varepsilon_j \sigma_j \rfloor j = 1, 2, \ldots, Q \tag{6}$$

where Q is the particle number total.

First, initialize the particles randomly and then update them iteratively. Three characters define each particle in iteration k.

(1) Location in search space $P_j(k)$

(2) The best position $P_{jbest}(k)$ at k iteration

(3) Flight speed $V_j(k)$

In addition, global optimal position of the entire particle swarm is defined as $P_{jbest}(k)$, so the function of velocity $V_j$ and position $P_j$ updated iteratively by each particle during flight is defined as:

$$\alpha(k) = (\alpha_{max} - \alpha_{min})(k/K)^2 + \alpha_{min} \tag{7}$$

$$c_1(k) = (c_{1max} - c_{1min})(k/K)^2 + c_{1min} \tag{8}$$

---

**Algorithm 1**

**Input:** m training samples: $\{(x_1, y_1), (x_2, y_2), \ldots, (x_m, y_m)\}$, activation function, loss function $J(\theta)$, network layers L, iteration number I and threshold $\varepsilon$ for the iteration stop;

**Output:** weight matrix $W$ and bias $b$

1. Initializing the weight matrix $\mathbf{W}$ and bias $b$ in the network layer

2. **for** $t = 1$ to I **do**

3. for i = 1 to m **do**

4. Initialize vector $a^1 = x_i$;

5.     **for** l=2 to L **do**

6.       $a^{i,l} = \sigma(z^{i,l}) = \sigma(W^l a^{i,l-1} + b^l)$;

7     **end for**

8. Calculate the gradient of parameters of each layer according to the loss function $\nabla_\theta J_t(\theta_{t-1})$

9. **end for**

10. **for** 1=2 to L **do**

11. $w^l := w^l - \alpha \cdot \nabla_W J_t(\theta_{t-1})^l$

12. $b^l := b^l - \alpha \cdot \nabla_b J_t(\theta_{t-1})^l$

13. **end for**

14. If the change value of all parameters is less than the iteration threshold $\epsilon$, then skip to step 15

15. **end for**

16. **return** weights matrix $W$ and bias vector $b$

---

$$c_2(k) = (c_{2\min} - c_{2\max})(k/K)^2 + c_{2\max} \tag{9}$$

$$v(k+1) = \alpha(k)v_j(k) + c_1(k)r_1\lfloor P_j(k) - P_{jbest}(k)\rfloor + c_2(k)r_2\lfloor P_j(k) - P_{gbest}(k)\rfloor \tag{10}$$

$$P_j(k+1) = P_j(k) + v_j(k+1) \tag{11}$$

In Formula (7) to (11), $\alpha(k)$ is the inertial velocity weight, $c_1(k)$ and $c_2(k)$ is the coefficient of acceleration, $r_1$ and $r_2$ are the independent random numbers located at $0 \sim 1$, K is the max iteration. The framework of the PSO-MLP is shown in Fig. 3.

## EXPERIMENTS AND ANALYSIS

Multimodal physiological data analysis can significantly improve the understanding and monitoring of student emotions and states in distance education. These data typically include, but are not limited to, various physiological signals such as heart rate, GSR, EEG, eye tracking, and facial expression. These data are collected by various sensor devices and transmitted in real-time to the analysis system for processing and interpretation. To verify the model using Python, a PSO-MLP neural network model was constructed, where preprocessed data was input for prediction. Additionally, a comparison was made with a MLP neural network.

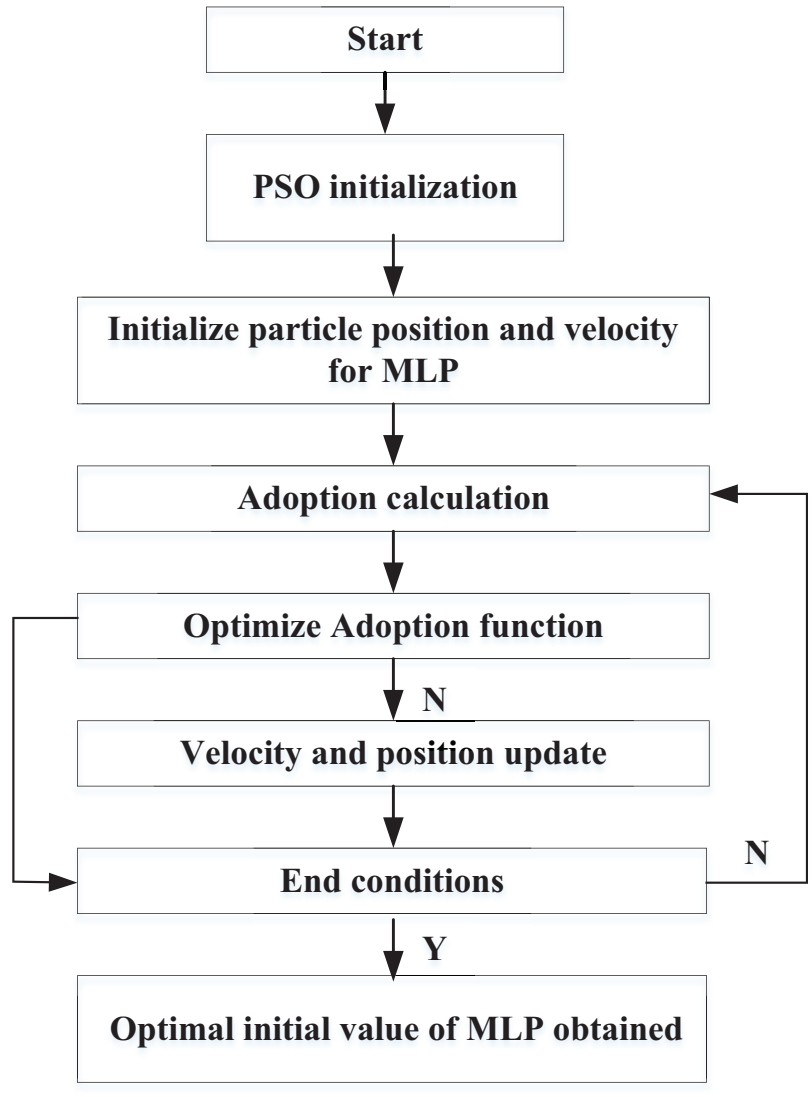

**Figure 3  Framework of PSO-MLP.**               

## Datasets

The DEAP (Dataset for Emotion Analysis using Physiological signals) dataset (https://paperswithcode.com/dataset/deap, DOI: 10.1109/T-AFFC.2011.15) was utilized to examine the emotional states of students based on their physiological responses while watching music videos. The dataset includes multimodal physiological signals from 32 participants, including EEG, GSR, heart rate, and facial expression data. Initially, the dataset was meticulously organized to ensure that the type of physiological signal systematically categorized each participant's data. This preparation phase was crucial for maintaining consistency throughout the analysis.

In the preprocessing stage, various techniques were applied to clean and normalize the data. For EEG data, a bandpass filter was used to remove noise and artifacts, followed by normalization to achieve zero mean and unit variance. The EEG signals were then

segmented into non-overlapping windows, typically one second long, to facilitate further analysis. Similarly, GSR, heart rate, and facial expression data were cleaned by removing outliers and artifacts, with segmentation performed in sync with the EEG data to maintain temporal alignment. We used the publicly available DEAP dataset, which contains EEG recordings from 32 participants across 32 video stimuli. To preprocess the data, the following steps were applied:

Filtering: EEG signals were bandpass filtered using a 4th-order Butterworth filter in the frequency range of 4–45 Hz to remove low-frequency drift and high-frequency noise.

Normalization: Each EEG channel was standardized using z-score normalization, computed across the entire signal of each channel per trial.

Segmentation: The continuous EEG signal was divided into overlapping segments using a 4-s window with 50% overlap, resulting in windows of 512 samples (at a sampling rate of 128 Hz).

Feature extraction: From each segment, we extracted statistical and frequency-domain features, including mean, standard deviation, skewness, kurtosis, and band power (delta, theta, alpha, beta, gamma bands).

Following preprocessing, feature extraction was performed to quantify and grade the physiological signals. For EEG data, features such as power spectral densities, band powers, and Hjorth parameters were extracted. Time-domain features, including mean, variance, standard deviation, skewness, and kurtosis, were derived from GSR and heart rate data. Sliding window techniques, with a window size of 5–10 s and a 50% overlap, were implemented to capture temporal changes in continuous signals, such as GSR and heart rate. These extracted features were statistically analyzed within each window to provide a comprehensive overview of the participants' physiological responses during the video sessions.

The next phase involved classifying and labeling the data based on the extracted features. The emotional states of students were categorized into three distinct groups: Active, Negative, and Exhausted. This classification was achieved using supervised machine learning algorithms trained on labeled data. The categorization allowed for a detailed analysis of each participant's class status, specifically identifying their levels of enthusiasm and fatigue. By mapping the physiological features to the defined class labels, the study was able to discern patterns in student engagement and emotional responses.

Finally, the processed data were analyzed to conclude the relevance to educational reform. The statistical analysis focused on understanding the correlation between physiological signals and class engagement levels, identifying patterns that indicated high enthusiasm, moderate participation, or signs of fatigue. These insights were then utilized to develop recommendations for optimizing class structures, teaching methods, or scheduling breaks to enhance student engagement and reduce fatigue. Cross-validation techniques were employed to assess the robustness of the classification model, with performance metrics such as accuracy, precision, recall, and F1-score being calculated.

**Table 1  Parameter settings.**

|  | Hyperparameter | Values |
|---|---|---|
| MLP architecture | Number of input neurons | 14 |
|  | Number of hidden layers | 2 |
|  | Number of output neurons | 1 |
|  | Hidden layer activation functions | ReLU |
| PSO optimization results | Number of neurons in the hidden layer | 78 |
|  | Dropout_rate | 0.14 |
|  | Batch_size | 95 |

## Experimental setup

All experiments were conducted using a Python-based implementation of the PSO-MLP model. The computational environment was configured on a workstation equipped with an Intel Core i7-12700K CPU (3.6 GHz, 12 cores), 32 GB of RAM, and an NVIDIA GeForce RTX 3080 GPU with 10 GB of VRAM, running Windows 11 Pro. Model training and evaluation were performed using Python 3.9, with key libraries including PyTorch 1.13, NumPy 1.23, sci-kit-learn 1.1, and SciPy 1.10. The optimization process, utilizing PSO, was implemented with custom modules and validated against benchmark results. All code was executed within the Anaconda environment, and experiments were repeated five times to ensure statistical stability.

Table 1 outlines the architecture hyperparameter values for MLP and the optimization results using PSO. For the experiments, the parameters were set as follows: MLP had an input neuron count of 14, two hidden layers, and a rectified linear unit (ReLU) activation function for the hidden layers.

In this experimental training, due to the high number of parameters and limited training samples, overfitting is prone to occur. Therefore, a Dropout layer is applied to the input to mitigate this issue. During training, each input neuron is retained with a probability $P$, while the others are temporarily deactivated. This approach ensures that the trained model does not overly rely on specific local features, thereby enhancing its generalization performance.

The selection of 14 input neurons is based on the dataset's input feature dimensionality, ensuring the network can adequately represent the input space. The choice of two hidden layers is motivated by the need for a balance between model complexity and training efficiency, with ReLU activation functions selected for their efficiency in mitigating vanishing gradient problems and facilitating faster convergence.

The Dropout_rate of 0.14 was determined empirically to strike a balance between preventing overfitting and maintaining model performance. A dropout rate that is too high could hinder the model's ability to learn effectively, while a rate that is too low might not sufficiently address overfitting.

The batch size of 95 was chosen to fully leverage GPU capabilities and improve training speed while maintaining the stability of gradient updates. The value of 95 was found to

provide a good trade-off between computational efficiency and convergence stability during preliminary experiments.

In the construction of the PSO-MLP model, the particle swarm algorithm optimizes parameters such as the number of neurons in the hidden layers, Dropout Rate, and Batch Size. The inertia weight $\omega = 0.5$, and the acceleration constants $c1 = c2 = 1$ are also set for PSO.

We evaluated model performance using accuracy, precision, recall, and F1-score. These metrics were calculated using the following formulas:

$$\text{Precision} = TP/(TP + FP) \tag{12}$$
$$\text{Recall} = TP/(TP + FN) \tag{13}$$
$$\text{F1-score} = 2 \times (\text{Precision} \times \text{Recall})/(\text{Precision} + \text{Recall}). \tag{14}$$

We applied five-fold cross-validation to ensure robustness. All metrics were averaged across the five folds. The same validation strategy was applied uniformly to both the baseline and proposed models for a fair comparison.

## Learning status recognition with different modal data

The identification was carried out based on the collected data the results are provided in Fig. 4.

According to the identification results using different modal data in Fig. 5, it can be observed that when single modal data is used for identification, the differences between the results of the three types of states are small, and the results of the two states in different states are not significant. However, when the survey information and health information are combined and input into the model simultaneously, the recognition rate improves to a certain extent. It can be seen that although achieving good results in a single mode is difficult, the advantages of MLP become apparent when information is combined.

To provide a more detailed explanation of the recognition process and to inform the subsequent education reform, we also analyzed the recognition rate at different times of day during the experiment. For the morning, afternoon and night, it is consistent with the overall results. The combined features have achieved a high recognition rate, with accuracy exceeding 90%. Overall, the recognition rate for the evening session is slightly lower than that of the other two time periods. This is because the students in the evening session, due to the matching of the day, had a large deviation when filling in the survey. This error was identified through discussions with the students themselves, suggesting that we can redistribute the weight of data from different modes when entering future research to ensure a high recognition rate. On this basis, to make a detailed analysis of the current education, we also analyzed the proportion of students in different learning states in three periods of the day, and the results are shown in Fig. 6.

The proportion of students in different periods, as shown in Fig. 6, is generally consistent with expectations: in the morning, students are more active, while those who are tired and negative account for a relatively small proportion. With the continuous increase in learning time per day, the proportion of students who are active decreases. However, it is worth noting that the proportion of students who are exhausted is the least at night, which

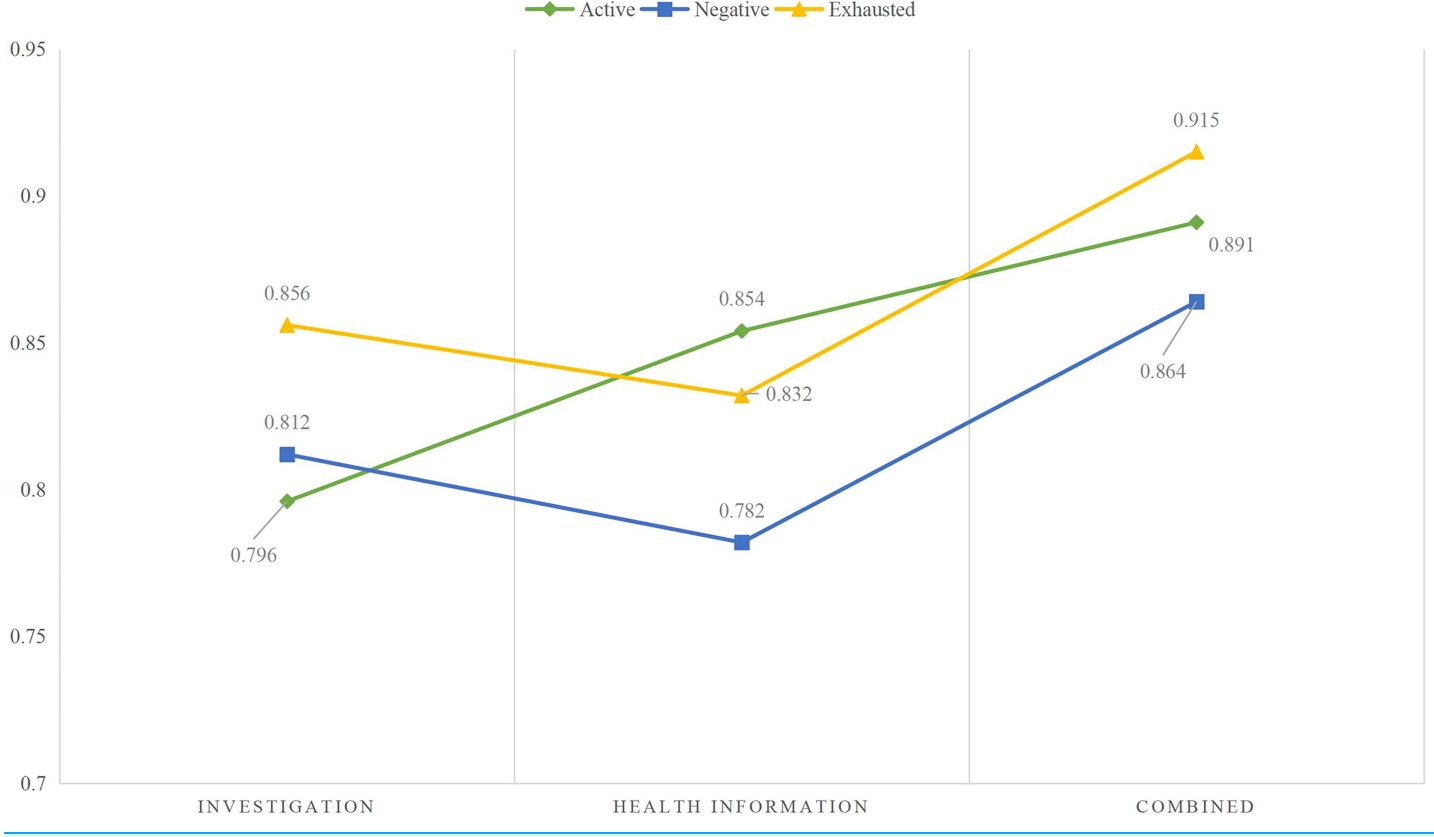

**Figure 4  The learning status recognition using different modal data.**               

deviates slightly from expectation. After communicating with students, it was found that most evening courses are elective or relatively low-difficulty courses. After a long rest in the evening, students' enthusiasm has recovered to a certain extent, resulting in the lowest proportion of exhausted courses. However, the proportion of active courses has increased compared to the noon courses.

Figure 7 presents the confusion matrix for the emotion recognition task. The results indicate that the method accurately identifies high arousal and moderate valence emotions, with accuracies of 90% and 94%, respectively. However, the accuracy for identifying low arousal and low valence emotions is lower, at only 61% and 58%, respectively. This discrepancy is due to the significant physiological changes typically associated with high arousal and high valence emotions, such as surprise and fear during stimulation, which leads to higher recognition accuracy. In contrast, the physiological changes associated with low arousal and low valence emotions are less pronounced, resulting in lower recognition accuracy.

## Comparison of the PSO-MLP and other methods

To verify the effectiveness of the proposed approach, we compared it with several existing methods, as shown in Fig. 8.

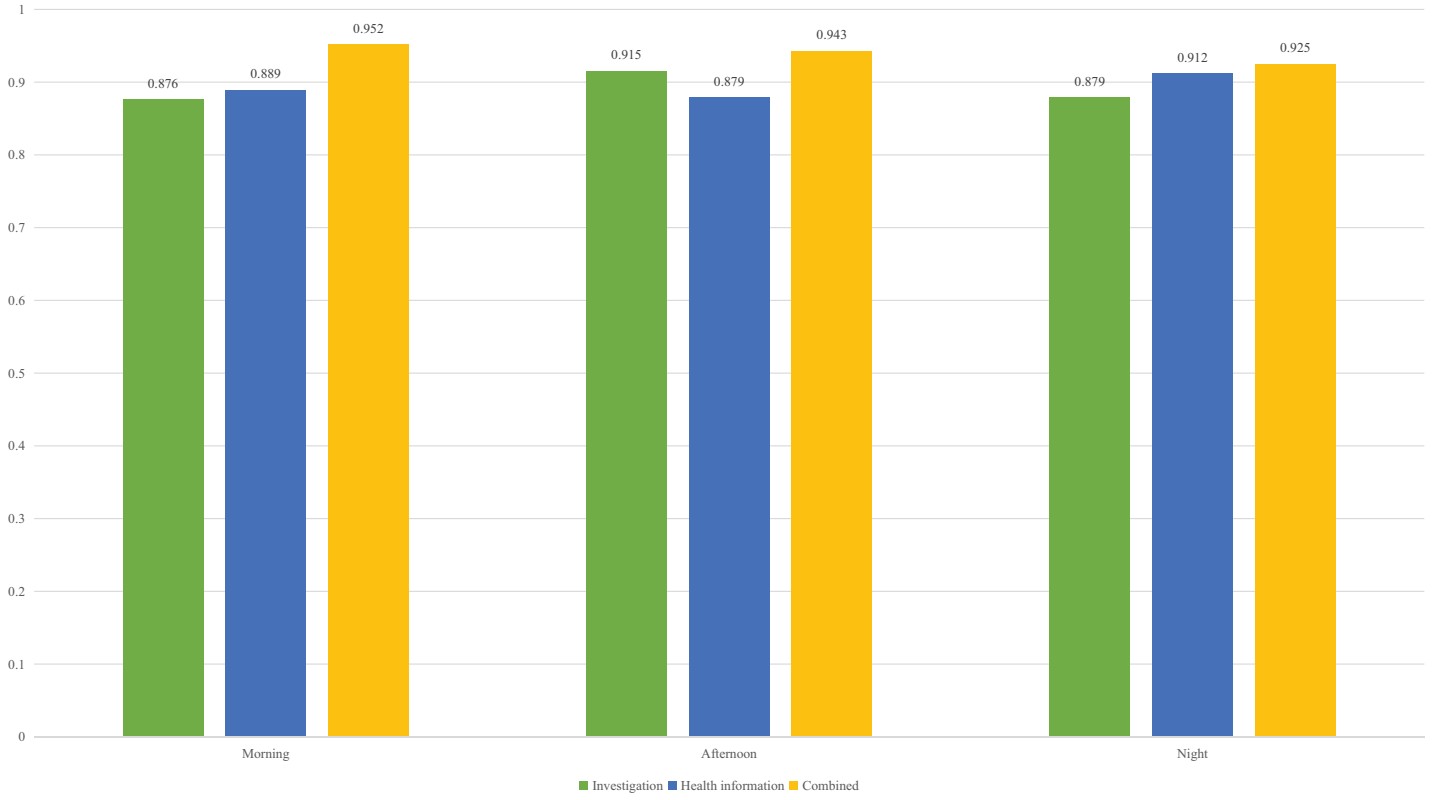

**Figure 5 The recognition accuracy in different times of the day.**

According to the comparison of different methods in Fig. 8, when the PSO method is not used for optimization, MLP does not have an advantage in the overall recognition rate. Its accuracy is lower than that of SVM and higher than that of the traditional NN method. The observed 0.029 accuracy gap between PSO combined with an MLP and SVM can be attributed to several underlying factors. Firstly, the inherent characteristics of the algorithms play a crucial role in their performance. PSO is a metaheuristic optimization technique inspired by swarm behavior and is used to optimize the weights and biases in MLPs. While MLPs can model complex patterns through their layered architecture, their performance is highly dependent on the tuning of hyperparameters and network structure.

In contrast, support vector machines (SVMs) are designed to find an optimal hyperplane that separates different classes and are generally effective in high-dimensional spaces. However, their performance can be sensitive to the choice of kernel function and its parameters. Table 2 summarizes the classification performance of the proposed PSO-MLP model compared to baseline models (MLP and SVM), including average metrics, standard deviations, 95% confidence intervals, and statistical significance (*p*-values).

However, with parameter optimization using PSO, the overall recognition rate has been further improved, as PSO helps the MLP avoid falling into local extreme values, thereby increasing the recognition accuracy even further.

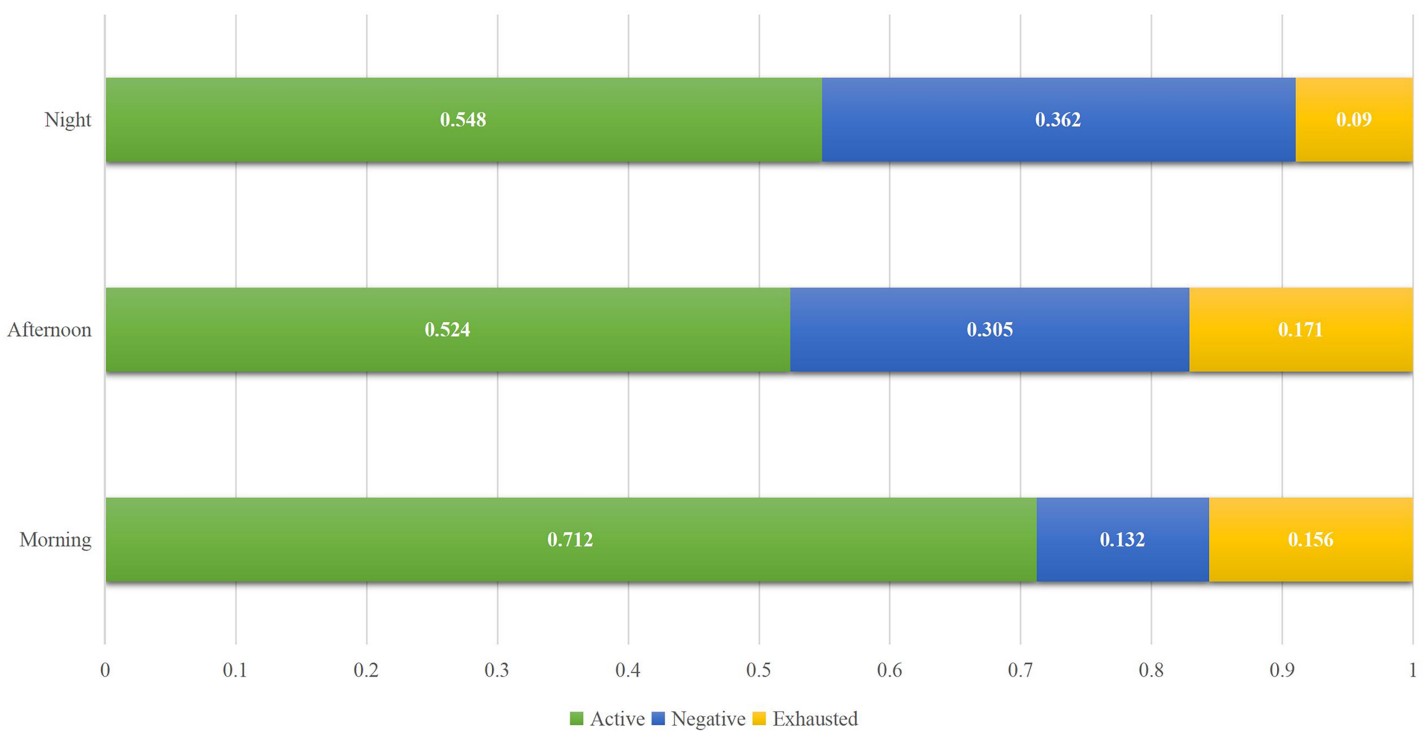

**Figure 6** Students' status ratio in different time.               

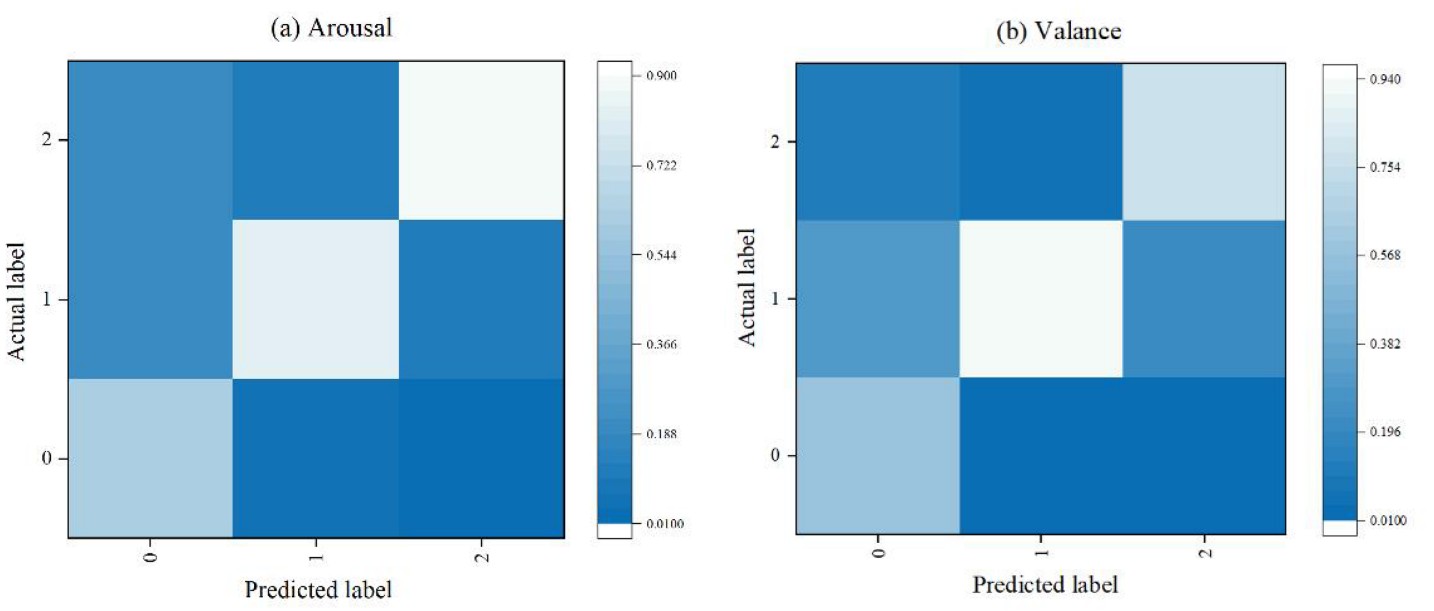

**Figure 7** Confusion matrix for distance learning status identification.     

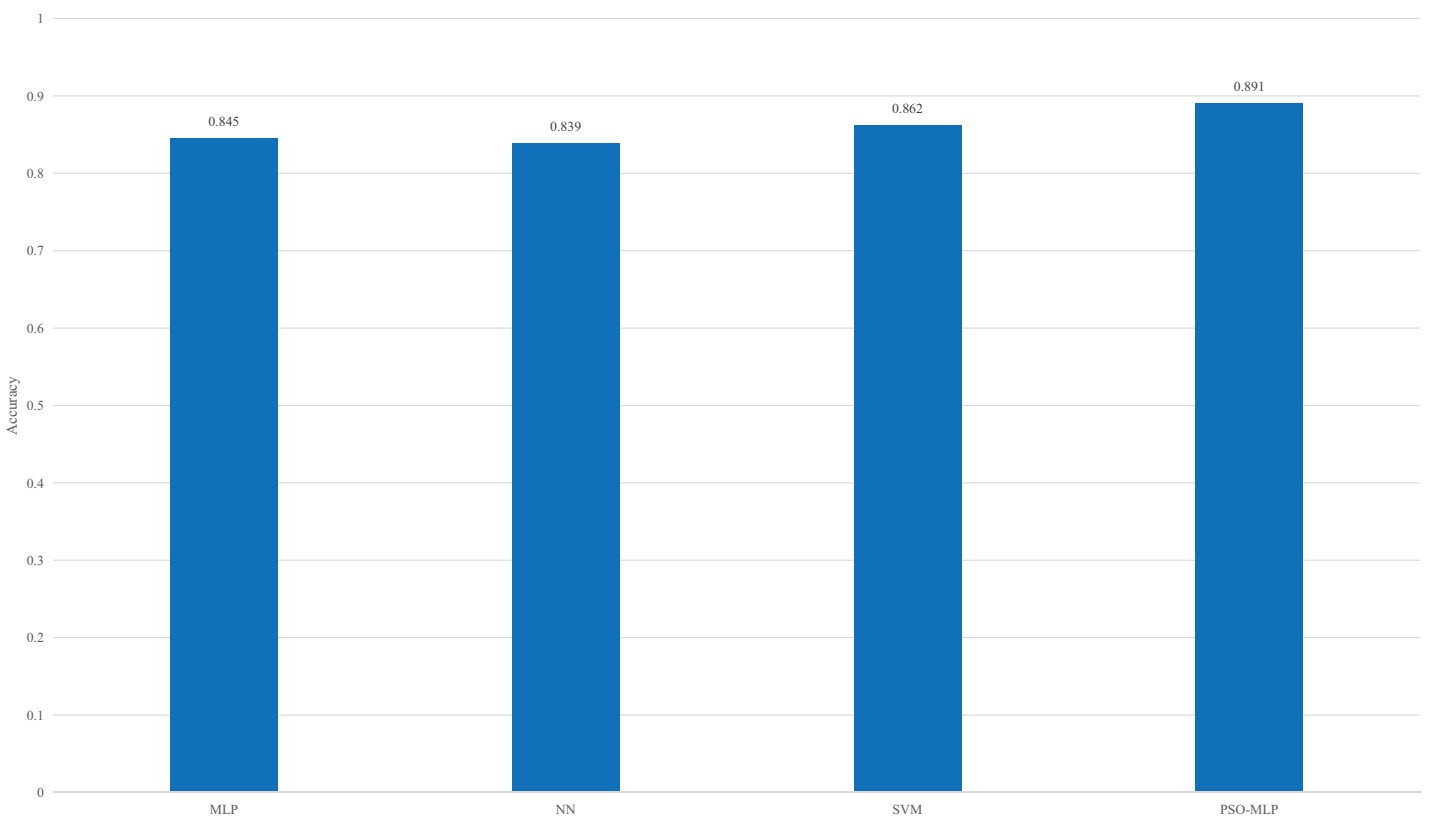

**Figure 8 Result of the comparison among different methods.**

**Table 2 Model performance comparison with statistical analysis.**

| Model | Accuracy (%) | Precision (%) | Recall (%) | F1-score (%) | 95% Confidence Interval (Accuracy) | *p*-value *vs.* PSO-MLP |
|---|---|---|---|---|---|---|
| PSO-MLP | 89.1 ± 1.3 | 88.7 ± 1.4 | 89.5 ± 1.1 | 89.1 ± 1.2 | [87.4–90.8] | – |
| MLP | 84.5 ± 1.6 | 83.9 ± 1.7 | 84.2 ± 1.8 | 84.0 ± 1.5 | [82.6–86.4] | 0.002 |
| SVM | 82.7 ± 1.9 | 81.8 ± 2.1 | 82.3 ± 2.0 | 82.0 ± 1.9 | [80.2–85.2] | 0.001 |

## DISCUSSION

This study addresses several specific challenges within the realm of remote education. Firstly, it introduces an intelligent method for assessing students' learning states by combining multimodal physiological data. By integrating real-time physiological signals such as heart rate, GSR, EEG, eye tracking, and facial expression, the study aims to provide educators with more precise insights into each student's emotional and cognitive states. This approach enables timely interventions and personalized support measures tailored to individual student needs, thereby enhancing the effectiveness of remote educational practices.

Remote education often struggles with accurately gauging student engagement and participation, making it difficult for educators to provide the necessary support. The use of MLP in this study to classify classroom engagement levels is a significant step forward. By

categorizing students based on their physiological data, educators can better identify which students are actively engaged and which may require additional motivation or support. This aspect of the research is crucial for fostering a more interactive and engaging learning environment, which is often challenging to achieve in online education contexts. Furthermore, the research provides essential technical support for educational health analysis. By employing AI-based tools, the study facilitates intelligent analysis of educators' and students' health statuses within educational settings. For instance, GSR can reflect emotional fluctuations, EEG analysis provides measurements of cognitive load and concentration, and eye tracking reveals the distribution of reading and attention. This technological advancement not only monitors but also improves the overall well-being of teachers and students, thereby contributing significantly to the broader goals of educational reform. The integration of such innovative approaches underscores the study's commitment to enhancing the quality and reliability of artificial intelligence (AI)-assisted educational systems in mobile and remote learning environments. Although the proposed model demonstrates promising results on the DEAP dataset, we acknowledge that relying solely on a single dataset collected in a controlled experimental setting limits the generalizability of the findings. DEAP reflects physiological responses to predefined video stimuli, which may not fully represent the complexity and variability of real-world educational environments. While we did not conduct external validation or domain adaptation in this study, we have provided a thorough discussion of these limitations. We emphasize the need for future work to explore the model's applicability in more diverse and authentic learning contexts, and we recognize that transferability across domains remains a critical challenge. This reflection ensures a realistic interpretation of the results and avoids overstating the conclusions.

The current stage of education demands leveraging the convenience brought by science and technology to improve and rapidly promote education reform. Addressing health issues among teachers and students during this reform, the study employs the MLP method to jointly identify the learning status of teachers and students based on multimodal physiological data. This provides robust support for ensuring the health of teachers and students in the future. The use of diverse physiological signals, such as heart rate and EEG, which belong to different data modalities, highlights the advantages of neural network methods in solving such complex problems.

Neural networks, like MLP, are particularly well-suited for handling diverse and nonlinear data inputs. The study chose MLP over traditional neural network methods due to its combination of feedforward and backpropagation mechanisms, which significantly enhance recognition efficiency. MLP's ability to manually adjust nodes further improves its efficiency. Additionally, the performance of the MLP method is optimized using PSO, as discussed in 'Experimental Setup', which ensures high precision and reliability. While MLP may not be the most outstanding method in intelligent classification algorithms, optimizing its parameters yields satisfactory results, providing a high-precision reference for future educational reforms. This optimized approach offers substantial data support for implementing education reforms, ensuring more effective and tailored educational practices in remote learning environments. By addressing the key challenges of

engagement and health monitoring, this study makes a significant contribution to the advancement of remote education.

In summary, while the study demonstrates promising results within a controlled experimental setting, we recognize the importance of critically examining both the practical and methodological aspects of our approach. The integration of physiological data into remote learning assessment is still an emerging area, and our use of a PSO-optimized MLP model represents a novel step toward intelligent emotion-aware systems. However, real-world deployment demands more than experimental performance—it requires interpretability, adaptability, and computational efficiency. We have reflected on these aspects through a more comprehensive discussion of limitations and potential biases. Furthermore, to enhance methodological transparency and support future replication, we provide access to high-level pseudocode and the project's source code. We believe these additions not only strengthen the study's credibility but also make a meaningful contribution to the ongoing conversation about scalable, data-driven approaches in remote education.

## CONCLUSIONS AND LIMITATIONS

### Conclusions

The integration of artificial intelligence (AI) in educational settings, particularly in mobile and online learning environments, has ushered in transformative opportunities. This study introduces an intelligent method for assessing students' learning states using multimodal physiological data, aiming to enhance the effectiveness and efficiency of remote education. By leveraging the joint characteristics of pre-class mental surveys and health big data from teachers and students, our proposed PSO-MLP model offers a robust solution for accurately classifying students' enthusiasm levels during classes. Through experiments utilizing the DEAP dataset and employing a PSO-optimized MLP neural network, we achieved a significant recognition accuracy of 89.1%, surpassing traditional methods. This approach not only identifies students' states (Active, Negative, Exhausted) based on physiological signals, such as EEG, GSR, and facial expressions, but also demonstrates the model's effectiveness across different times of the day, with peak recognition rates exceeding 90%. Moreover, the study compares our PSO-MLP model against other methodologies, showcasing superior performance in recognition rates due to PSO's optimization capabilities, which mitigate local minima issues inherent in MLP training. This optimization ensures the robustness and applicability of our model in real-world educational contexts, supporting personalized teaching approaches and effective student health management.

Furthermore, our findings contribute essential insights into the challenges of remote education, particularly in gauging student engagement and well-being. By integrating advanced AI tools, such as MLP, enhanced by PSO, educators can make informed decisions to foster more interactive and supportive learning environments. This technological integration not only monitors but also enhances the overall well-being of both educators and students, thereby catalyzing educational reform efforts.

## Limitations

While the proposed method demonstrates promising performance, several important limitations must be acknowledged to present a more balanced assessment. First, the model's effectiveness relies heavily on the quality, representativeness, and diversity of the physiological data used. Although the DEAP dataset provides a solid foundation, it is collected under controlled experimental conditions. It may not fully capture the complexity, variability, and contextual diversity of real-world educational environments. This constraint limits the generalizability of the findings, especially when considering diverse student populations, varying emotional baselines, and environmental factors. Second, although the PSO-MLP model achieves high accuracy, its complexity introduces challenges related to interpretability. Neural networks, particularly those optimized through metaheuristic algorithms like PSO, often behave as "black boxes," making it difficult for educators or researchers to understand how predictions are made. Without clearer insights into feature importance or decision pathways, the practical application of the model in education may be hindered, as stakeholders may lack the necessary transparency to trust or act on the model's outputs.

Furthermore, the risk of overfitting must be considered. While cross-validation helps mitigate this, relying on a single dataset raises the possibility that the model may have adapted too closely to dataset-specific patterns rather than learning generalizable features. Additionally, there is a limited exploration of how performance varies across demographic subgroups or recording sessions, which could introduce unintended biases or instability. Finally, the deployment of such a system in real-time educational settings presents practical challenges. Real-time processing of multimodal physiological signals requires substantial computational resources, and system latency can compromise responsiveness and usability in dynamic classroom environments. These limitations underscore the importance of future work involving external validation, interpretability enhancement, and domain adaptation to bridge the gap between experimental performance and real-world utility.

### Funding

The authors received no funding for this work.

### Competing Interests

Muhammad Asif is an Academic Editor for PeerJ.

### Author Contributions

- Jing Wang conceived and designed the experiments, performed the experiments, analyzed the data, performed the computation work, prepared figures and/or tables, authored or reviewed drafts of the article, and approved the final draft.

• Muhammad Asif conceived and designed the experiments, analyzed the data, prepared figures and/or tables, authored or reviewed drafts of the article, and approved the final draft.

## Data Availability

The DEAP dataset is available at: https://paperswithcode.com/dataset/deap.

## Supplemental Information

Supplemental information for this article can be found online at http://dx.doi.org/10.7717/peerj-cs.3121#supplemental-information.

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
