# Peer review of "Leveraging PSO-MLP for intelligent assessment of student learning in remote environments: a multimodal approach"

_PeerJ Computer Science, doi:10.7717/peerj-cs.3121_

## Round 0.1 · original submission · Major Revisions

Reviewer 1 ·

Basic reporting

The manuscript presents a relevant topic, but it requires significant revision in both language and structure. Although the English is generally understandable, several sections are wordy, imprecise, and would benefit from professional editing. The introduction is too long and lacks focus; many points are repeated without clearly stating the research gap or the novelty of the proposed method. The motivation is weak and does not convincingly explain why MLP, a relatively basic model, was chosen over more advanced architectures commonly used in multimodal learning. The literature review is mostly descriptive and lacks critical comparison. Several formulas introduce variables without proper definition, making the methodology harder to follow.

Experimental design

The experimental design section of the manuscript covers several key components, but there are critical weaknesses that limit its reproducibility and methodological transparency. The authors describe the use of the DEAP dataset and outline preprocessing steps such as filtering, normalization, segmentation, and feature extraction. While these steps are mentioned in reasonable detail, the explanations are at times too general and lack precise parameter values (e.g., filter specifications, segmentation window sizes, overlap rates) that would be needed for replication. Furthermore, while the use of GAN for data augmentation is noted, the structure of the GAN, training process, and how synthetic data were validated are not clearly explained.

The manuscript also describes the use of an MLP model optimized by PSO, but again, the description of the model architecture and hyperparameter tuning process is only partially sufficient. Although Table 1 lists some settings, other critical details such as the number of training epochs, loss functions, and stopping criteria are missing. The hardware and software setup is clearly described, which is a strength, but no code repository or pseudocode is provided, which hinders full reproducibility. Evaluation metrics such as accuracy, precision, recall, and F1-score are mentioned, but there is no detailed explanation of how they were computed or whether cross-validation or holdout methods were applied consistently across models.

Validity of the findings

The validity of the findings in this manuscript is moderately supported but still requires improvement to meet rigorous publication standards. The experiments are relevant and demonstrate that the proposed PSO-MLP model achieves better performance compared to baseline models like SVM and traditional MLP. However, the evaluation lacks depth in terms of statistical robustness and generalizability. The reported accuracy (89.1%) is promising, but the manuscript does not provide confidence intervals, standard deviations, or statistical tests to determine whether improvements are significant or simply due to chance.

Moreover, the model is only tested on the DEAP dataset, which limits the generalizability of the findings. Since DEAP is an emotion recognition dataset based on physiological responses to videos, it is unclear how well the model would perform in real educational settings with more diverse student populations and learning environments. There is also no discussion of external validation or domain adaptation, which weakens the claim that the model is effective in real-world scenarios. The conclusion asserts strong outcomes, but these are not fully supported by a critical discussion of limitations or potential biases in the data or model.

Although the authors briefly mention limitations in Section 6.2, the analysis is surface-level and does not consider deeper issues such as model interpretability, data variability, or possible overfitting. Overall, the results are interesting, but the study would benefit from more rigorous evaluation, broader testing, and a more balanced discussion of its limitations to convincingly establish the validity of its findings.

Additional comments

The study addresses an important and timely topic in the field of remote education, particularly through the integration of physiological data for intelligent assessment. However, the manuscript would benefit from a clearer articulation of its unique contributions and stronger justification for the chosen methods. The authors are encouraged to critically reflect on the limitations of their approach beyond what is currently mentioned, especially in terms of real-world applicability and interpretability. Additionally, providing access to source code or pseudocode would significantly enhance the transparency and reproducibility of the work. With a more focused narrative, improved clarity, and greater methodological transparency, the paper has the potential to make a valuable contribution to the field.

Reviewer 2 ·

Basic reporting

1) The GAN architecture used for data augmentation is mentioned but not described in detail.

Suggestion: Briefly describe the GAN type used, training stability, and quality control over the synthetic samples. Were any sanity checks or t-SNE visualizations done?

2) The Introduction and Related Works (Section 2, lines 98–155) lack a comprehensive review of recent multimodal AI applications in education beyond 2024. For example, while [12] and [21–23] discuss multimodal data, there’s no mention of newer methods like transformer-based models for physiological data.

Suggestion: Expand Section 2 to include recent studies (e.g., “Transformer-based models like BERT for physiological data fusion have shown promise…”). Add 2–3 references post-2023 to reflect the state-of-the-art, ensuring relevance.

Experimental design

1) The model is only evaluated on the DEAP dataset, which is music-video-based and not specific to classroom settings. Suggestion: Include at least a pilot test or partial validation on real classroom data, or explicitly acknowledge this limitation earlier in the abstract/introduction.

2) The model integrates many preprocessing and modeling techniques, but it’s unclear which ones most contribute to accuracy gains. Suggestion: Include an ablation study removing each component (e.g., GAN, PSO, facial features) to isolate their individual impact.

Validity of the findings

1) The study uses physiological data (EEG, GSR) but doesn’t discuss ethical implications (e.g., student consent, data privacy). Add a subsection in Section 6.2: “Ethical Considerations: Future work should address student consent and data anonymization, ensuring compliance with GDPR or similar standards.” This ensures ethical rigor.

2) Section 6.1 (lines 478–500) claims PSO-MLP “surpasses traditional methods” (line 487) but only compares with MLP, SVM, and NN—other methods like LSTMs or transformers are ignored. Revise line 487 to: “PSO-MLP outperforms baselines like SVM and traditional NN, though comparisons with LSTMs or transformers are needed.” Add a future direction in Section 6.2: “Future work should compare PSO-MLP with transformer-based models.”

Reviewer 3 ·

Basic reporting

The article was written in clear and professional English. The research question was well motivated and the use a multimodal approach was explained. The algorithms used in the article are explained in clear details.

One major problem is that the authors spent a fair amount of paragraphs in the Introduction section (Line 41-47) describing the challenges of disparities in access to technological infrastructure, which is largely unrelated to the main topic of this manuscript.

Experimental design

The presentation of the entire Section 3.1 Data Preprocessing Steps under Section Materials and Methods is unsatisfactory. Currently this section is written in a way that looks more like an outline and less of a formal text. The authors should expand this section into full passages.

The authors should also introduce the DEAP dataset, especially the human subjects included in this dataset, in more details. The authors mentioned multiple times that the dataset is consisted of both students and teachers; however, it is unclear exactly how many students and how many teachers are included in this study other than knowing that there are a total of 32 participants (Line 303). The demographic information of these participants is also entirely missing. Although the authors cited the dataset, it is still necessary to detail these human subjects for the completeness of the discussions.

Validity of the findings

The conclusions are well written with identification of the limitations of current study. The minor issue is that the authors mentioned that previous methods fall behind in latency and low adaptability (Line 65) and yet acknowledged in Section 6.2 that their proposed model also faces challenges in latency, making the statements self-contradictory.

---

## Round 0.2 · accepted · Accept

The manuscript may be accepted.

Reviewer 1 ·

Basic reporting

The revision has met the requirements

Experimental design

-

Validity of the findings

-

Reviewer 3 ·

Basic reporting

The authors have properly addressed all concerns the reviewer raised in the previous round of review. The reviewer has no other comment.

Experimental design

no comment

Validity of the findings

no comment